# Clip Your Sequences Fairly: Enforcing Length Fairness for Sequence-Level RL

## Abstract

We propose **FSPO** (Fair Sequence Policy Optimization), a sequence-level reinforcement learning method for LLMs that enforces length-fair clipping on the importance-sampling (IS) weight. We study RL methods with sequence-level IS and identify a mismatch when PPO/GRPO-style clipping is transplanted to sequences: a fixed clip range systematically reweights short vs. long responses, distorting the optimization direction. FSPO introduces a simple remedy: we clip the sequence log-IS ratio with a band that scales as $\sqrt{L}$. Theoretically, we formalize length fairness via a Length Reweighting Error (LRE) and prove that small LRE yields a cosine directional guarantee between the clipped and true updates. Empirically, FSPO flattens clip rates across length bins, stabilizes training, and outperforms baselines across model sizes and evaluation datasets, with the largest gains on the Qwen3-8B-Base model.

## 1 Introduction

Recent progress on reinforcement learning (RL) for large language models (LLMs) has been catalyzed by GRPO (Shao et al., 2024) and the broader RLVR paradigm (Lambert et al., 2025), where rule-based, verifiable rewards are assigned to the entire response rather than token-wise signals. This framing has proven effective for improving mathematical reasoning and other verifiable tasks (DeepSeek-AI et al., 2025; Wen et al., 2025; Wang et al., 2025b). However, the optimization procedures used in current RLVR systems largely inherit token-level machinery from PPO-like methods (Schulman et al., 2017), including the use of token-level importance-sampling (IS) ratios and token-level clipping. Meanwhile, subsequent works emphasize that once rewards are sequence-level, it is more faithful to operate with sequence-level IS so as to match the reward granularity (Ahmadian et al., 2024; Zheng et al., 2025).

Despite the shift toward sequence-level IS, the theoretical distinctions and practical consequences of clipping in this setting remain underexplored. Existing sequence-level IS methods (Ahmadian et al., 2024; Zheng et al., 2025) transplant the clipping mechanism from token-level methods directly and apply a *fixed* clip range to the probability ratio of the whole sequence. We argue that fixed sequence-level clipping is problematic: the dispersion of sequence-level *log* ratios increases with response length $L$. A fixed band therefore induces length-dependent acceptance rates and systematically reweights short versus long responses.

This paper studies sequence-level clipping through the lens of *length fairness*. We formalize a simple criterion: *acceptance rates should be approximately constant across response lengths*. We show that fixed sequence-level clipping violates this criterion and can distort the training target. To address this, we propose **FSPO** (Fair Sequence Policy Optimization). FSPO preserves IS semantics and restores length fairness by using a $\sqrt{L}$-scaled acceptance band on the sequence log-ratio, which approximately equalizes acceptance across lengths.

To ground our analysis, we evaluate **FSPO** on sequence-level RL for mathematical reasoning. We compare against two sequence-level baselines: (i) *RLOO* with sequence-level IS and a fixed clip on the full-sequence ratio, and (ii) *GSPO* with ratio normalization. We report Avg@8 on MATH500, Avg@32 on AIME24 and AIME25, alongside diagnostic plots that measure acceptance rate as a

function of response length to verify length fairness. Across two base model scales, we observe flatter acceptance across length bins, more stable training dynamics, and improved task scores; full results and ablations are presented in Section 7.

Background on RL for LLMs and RLVR is provided in Section 2, and a detailed justification for sequence-level IS weights in RLVR scenario is given in Appendix A.

## 2 PRELIMINARIES AND RELATED WORK

**Setup.** Let $\boldsymbol{x}_i \in \mathcal{X}$ be a context (prompt) drawn from a data distribution $p(\boldsymbol{x})$, and let $\boldsymbol{y}_i = (y_{i,1}, \ldots, y_{i,|\boldsymbol{y}_i|}) \in \mathcal{Y}$ be a response (a token sequence). Under a policy $\pi_{\boldsymbol{\theta}}$ (an LLM in our setting), the sequence probability factorizes as

$$\pi_{\boldsymbol{\theta}}(\boldsymbol{y}_i \mid \boldsymbol{x}_i) = \prod_{t=1}^{|\boldsymbol{y}_i|} \pi_{\boldsymbol{\theta}}(y_{i,t} \mid \boldsymbol{h}_{i,t}),$$

where $\boldsymbol{h}_{i,t} = (\boldsymbol{x}_i, \boldsymbol{y}_{i,<t})$ is the token prefix at step $t$.

**RLHF and PPO.** Reinforcement Learning from Human Feedback (RLHF) (Ouyang et al., 2022) frames alignment as policy optimization against a reward model learned from human preference data. RLHF pipelines typically use PPO (Schulman et al., 2017): reward is given to the final token in a sequence and propagated to other tokens via GAE (Schulman et al., 2015b) to obtain per-token advantages $\hat{A}_{i,t}$. The standard PPO surrogate is

$$\mathcal{J}_{\text{PPO}}(\boldsymbol{\theta}) = \frac{1}{N} \sum_{i=1}^{N} \frac{1}{|\boldsymbol{y}_i|} \sum_{t=1}^{|\boldsymbol{y}_i|} \min\Big(r_{i,t}(\boldsymbol{\theta})\, \hat{A}_{i,t},\ \text{clip}\big(r_{i,t}(\boldsymbol{\theta}),\, 1-\epsilon,\, 1+\epsilon\big)\, \hat{A}_{i,t}\Big), \quad (1)$$

where $r_{i,t}(\boldsymbol{\theta}) = \dfrac{\pi_{\boldsymbol{\theta}}(y_{i,t} \mid \boldsymbol{h}_{i,t})}{\pi_{\boldsymbol{\theta}_{\text{old}}}(y_{i,t} \mid \boldsymbol{h}_{i,t})}$ is the token-level IS ratio.

**RLVR paradigm and GRPO.** For math, programming, and other verifiable tasks, recent systems adopt the RLVR paradigm (Lambert et al., 2025; Wen et al., 2025), applying rule-based, sequence-level rewards that can be automatically checked. Representative methods include GRPO (Shao et al., 2024), DAPO (Yu et al., 2025), DrGRPO (Liu et al., 2025), etc. A typical GRPO-style update draws a group of $G$ completions $\{\boldsymbol{y}_i\}_{i=1}^{G}$ for the same prompt $\boldsymbol{x}$ under $\pi_{\boldsymbol{\theta}_{\text{old}}}$ (here the index $i$ is reused as the within-group index; earlier $i$ indexed dataset examples), computes sequence rewards $R_i = G(\boldsymbol{x}, \boldsymbol{y}_i)$, and uses the group mean as a baseline so that $\hat{A}_i = \big(R_i - \frac{1}{G}\sum_{j=1}^{G} R_j\big)/\sigma$, where $\sigma$ is the within-group reward standard deviation. The token advantages set $\hat{A}_{i,t} = \hat{A}_i$ for all $t$. The GRPO objective mirrors equation 1 and therefore inherits token-level ratios and clipping.

**Sequence-level importance sampling.** A growing line of work argues that when rewards are sequence-level, policy optimization should use sequence-level IS. RLOO (Ahmadian et al., 2024) models the LLM as a one-step bandit and treats an entire response as an action. According to the TRL implementation `RLOO_Trainer` (Hugging Face TRL Team, 2025), the objective is

$$\mathcal{J}_{\text{RLOO}}(\boldsymbol{\theta}) = \mathbb{E}_{\boldsymbol{x},\, \{\boldsymbol{y}_i\} \sim \pi_{\boldsymbol{\theta}_{\text{old}}}} \left[ \frac{1}{G} \sum_{i=1}^{G} \min\Big(s_i(\boldsymbol{\theta})\, \hat{A}_i^{\text{LOO}},\ \text{clip}\big(s_i(\boldsymbol{\theta}),\, 1-\epsilon,\, 1+\epsilon\big)\, \hat{A}_i^{\text{LOO}}\Big) \right], \quad (2)$$

where

$$s_i(\boldsymbol{\theta}) = \frac{\pi_{\boldsymbol{\theta}}(\boldsymbol{y}_i \mid \boldsymbol{x})}{\pi_{\boldsymbol{\theta}_{\text{old}}}(\boldsymbol{y}_i \mid \boldsymbol{x})} \quad (3)$$

is the *sequence-level* IS ratio that matches reward granularity, and $\hat{A}_i^{\text{LOO}}$ is the leave-one-out unbiased estimator in Kool et al. (2019). GSPO (Zheng et al., 2025) pursues the same goal but normalizes the ratio by length (e.g., $s_i^{\text{norm}} = \exp(\frac{1}{|\boldsymbol{y}_i|} \log s_i)$) before clipping:

$$\mathcal{J}_{\text{GSPO}}(\boldsymbol{\theta}) = \mathbb{E}\left[ \frac{1}{G} \sum_{i=1}^{G} \min\Big(s_i^{\text{norm}}(\boldsymbol{\theta})\, \hat{A}_i,\ \text{clip}\big(s_i^{\text{norm}}(\boldsymbol{\theta}),\, 1-\epsilon,\, 1+\epsilon\big)\, \hat{A}_i\Big) \right]. \quad (4)$$

While length normalization aims to equalize scales across different lengths (Zheng et al., 2025), we argue that it does not actually balance the clipping scale and also undermines IS correctness.

## 3 LENGTH FAIRNESS AND LENGTH REWEIGHTING ERROR (LRE)

Following the notation in Section 2, define the sequence-level log importance ratio

$$S(\boldsymbol{y} \mid \boldsymbol{x}) = \log \pi_{\boldsymbol{\theta}}(\boldsymbol{y} \mid \boldsymbol{x}) - \log \pi_{\boldsymbol{\theta}_{\text{old}}}(\boldsymbol{y} \mid \boldsymbol{x}).$$

Let $L = \text{len}(\boldsymbol{y})$ and fix a length-indexed band $b_L > 0$. The acceptance event (unclipped) is defined as

$$\mathcal{A}_L = \{\, \boldsymbol{y} \mid |S(\boldsymbol{y} \mid \boldsymbol{x})| \le b_L \,\}.$$

We define the length-conditional acceptance rate $q(L) = \text{Pr}_{\pi_{\boldsymbol{\theta}_{\text{old}}}}(\mathcal{A}_L \mid L)$, and the per-length contributions

$$\boldsymbol{g}_L^{\star} = \mathbb{E}_{\pi_{\boldsymbol{\theta}_{\text{old}}}} \Big[ \frac{\pi_{\boldsymbol{\theta}}(\boldsymbol{y} \mid \boldsymbol{x})}{\pi_{\boldsymbol{\theta}_{\text{old}}}(\boldsymbol{y} \mid \boldsymbol{x})} \nabla_{\boldsymbol{\theta}} \log \pi_{\boldsymbol{\theta}}(\boldsymbol{y} \mid \boldsymbol{x}) \, A(\boldsymbol{x}, \boldsymbol{y}) \,\Big|\, L \Big],$$

$$\boldsymbol{g}_L^{b} = \mathbb{E}_{\pi_{\boldsymbol{\theta}_{\text{old}}}} \Big[ \frac{\pi_{\boldsymbol{\theta}}(\boldsymbol{y} \mid \boldsymbol{x})}{\pi_{\boldsymbol{\theta}_{\text{old}}}(\boldsymbol{y} \mid \boldsymbol{x})} \nabla_{\boldsymbol{\theta}} \log \pi_{\boldsymbol{\theta}}(\boldsymbol{y} \mid \boldsymbol{x}) \, A(\boldsymbol{x}, \boldsymbol{y}) \,\Big|\, \mathcal{A}_L, \, L \Big],$$

so that the true policy gradient target and its clipped surrogate are

$$\boldsymbol{g}^{\star} = \mathbb{E}_L\big[\boldsymbol{g}_L^{\star}\big], \qquad \boldsymbol{g}^{b} = \mathbb{E}_L\big[q(L)\,\boldsymbol{g}_L^{b}\big].$$

**Definition 3.1** (Length Reweighting Error (LRE)). *Let $\bar{q} = \mathbb{E}\big[q(L)\big]$. Define*

$$\text{LRE} = \tfrac{1}{2} \mathbb{E}\bigg[\bigg| \frac{q(L)}{\bar{q}} - 1 \bigg|\bigg].$$

Small LRE means the acceptance rate is nearly constant across response lengths.

Let $\kappa = \dfrac{\mathbb{E}\big[\|\boldsymbol{g}_L^{\star}\|\big]}{\|\boldsymbol{g}^{\star}\|} \ge 1$, which captures the dispersion of per-length signal magnitude.

**Assumption 3.1** (Bounded stratification). *There exists $\eta \in [0, 1)$ such that for all $L$,*

$$\big\|\boldsymbol{g}_L^{b} - \boldsymbol{g}_L^{\star}\big\| \le \eta \, \|\boldsymbol{g}_L^{\star}\|.$$

This assumption states that clipping does not severely distort the target within each length stratum.

**Assumption 3.2** (Bounded correlation). *The correlation between $|q(L) - \bar{q}|$ and $\|\boldsymbol{g}_L^{\star}\|$ is mild so that*

$$\mathbb{E}\Big[\, |q(L) - \bar{q}| \, \|\boldsymbol{g}_L^{\star}\| \,\Big] \le \gamma \, \mathbb{E}\big[|q(L) - \bar{q}|\big] \, \mathbb{E}\big[\|\boldsymbol{g}_L^{\star}\|\big].$$

This assumption is optional; see Appendix B.

**Theorem 3.1** (Directional guarantee under length fairness). *Under Assumptions 3.1 and 3.2,*

$$\cos \angle\big(\boldsymbol{g}^{b}, \boldsymbol{g}^{\star}\big) \ge \frac{1 - \rho}{1 + \rho}, \qquad \rho \le \kappa\Big(\eta + 2\gamma(1 + \eta)\,\text{LRE}\Big).$$

The theorem implies that smaller LRE yields a larger lower bound on the cosine similarity between the clipped update and the true update. The proof and further discussion are provided in Appendix B.

## 4 DISTRIBUTION OF SEQUENCE-LEVEL LOG RATIO

In this section we study the distribution of the sequence-level log importance-sampling (IS) ratio and derive practical guidance for designing procedures that achieve the *length fairness* criterion introduced earlier.

We view decoding under an LLM $\pi$ with a limited context window $K$ as a finite-state Markov chain on $V^K$, where $V$ is the vocabulary; this reduction for autoregressive language models is discussed by Zekri et al. (2025) in detail. Under randomized sampling with nonzero temperature, the chain

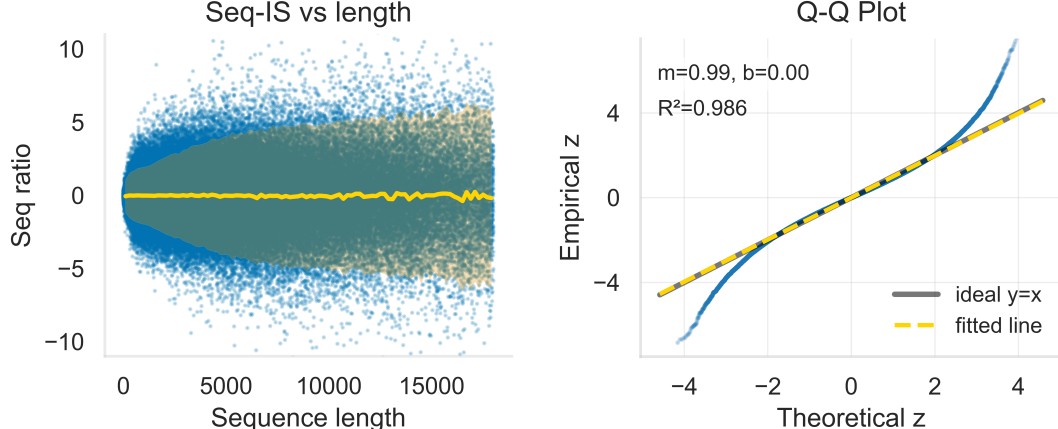

Figure 1: **Empirical analysis of the sequence-level IS ratio.** Sample size $n = 217{,}454$. **Left:** Empirical distribution of $S_L$. The yellow line shows the empirical mean and the shaded band the $\pm 2$ empirical standard deviation, computed with a bin size of 200 (see Appendix B for justification of binning). **Right:** Q–Q plot testing normality. The sorted data point quantiles are shown in blue dots. We report the slope $m$, intercept $b$, and $R^2$ of the fitted line.

is irreducible and aperiodic. Therefore, the Markov CLT for additive functionals (Jones, 2004; Maxwell & Woodroofe, 2000) applies to

$$S_L = \sum_{t \leq L} \log \frac{\pi_{\boldsymbol{\theta}}(\boldsymbol{y}_t \mid \boldsymbol{h}_t)}{\pi_{\boldsymbol{\theta}_{\text{old}}}(\boldsymbol{y}_t \mid \boldsymbol{h}_t)},$$

which yields the following theorem:

**Theorem 4.1** (Gaussianity of the sequence-level log ratio). *The sequence log-IS ratio obeys an asymptotically Gaussian law:*

$$\frac{S_L - \mu_L}{\sqrt{L}} \Rightarrow \mathcal{N}(0, \sigma^2), \qquad \mu_L = \Theta(L), \quad \sigma^2 > 0.$$

Figure 1 illustrates the empirical distribution of the sequence-level log IS ratio using all steps across the full training run. Consistent with the theorem, the empirical standard deviation of $S_L$ grows approximately $\propto \sqrt{L}$. The observed estimator is $\hat{\sigma} = 0.0304$.

To further assess normality, we compute the standardized statistic

$$\hat{Z} = \frac{S_L - \hat{\mu}_L}{\sqrt{L}\,\hat{\sigma}},$$

where $\hat{\mu}_L$ is computed within each length bin and $\hat{\sigma}$ is estimated from all values of $(S_L - \hat{\mu}_L)/\sqrt{L}$. The Q–Q plot (right panel) shows that the fitted line coincides with the $y = x$ reference; the empirical distribution exhibits slightly heavier tails, but within $\pm 2$ standard deviations it is very close to normal.

In Figure 1 (left), the estimated per-length mean $\hat{\mu}_L$ exhibits slightly larger fluctuations at larger lengths but remains small relative to $\hat{\sigma}$, thus empirically we set $\hat{\mu}_L \approx 0$.

**Theoretical clip-fraction patterns of RLOO and GSPO.** By Theorem 4.1, $S_L \approx \mathcal{N}(\mu_L, \sigma^2 L)$. Let $\Phi(\cdot)$ be the standard normal CDF. Similar to the acceptance-rate notation $q(L)$ used in Section 3, We denote the *clip probability* by $c(L) := 1 - q(L)$. For a symmetric two-sided clip in log space:

$$\textbf{RLOO:} \quad c_{\text{RLOO}}(L) = \Pr\big(|S_L| > \xi\big) = 2\,\Phi\left(-\frac{\xi - \mu_L}{\sigma\sqrt{L}}\right) \approx 2\,\Phi\left(-\frac{\xi}{\sigma\sqrt{L}}\right), \tag{5}$$

$$\textbf{GSPO:} \quad c_{\text{GSPO}}(L) = \Pr\big(|S_L| > \xi L\big) = 2\,\Phi\left(-\frac{\xi L - \mu_L}{\sigma\sqrt{L}}\right) \approx 2\,\Phi\left(-\frac{\xi\sqrt{L}}{\sigma}\right), \tag{6}$$

where the approximations use the empirically small drift $\mu_L \approx 0$. Both schemes induce clip probabilities that vary systematically with $L$.

To obtain a *constant* (length-independent) clip probability, **FSPO** sets $b_L = \mu_L + z\,\sigma\,\sqrt{L}$. With the same calculation as in Equations (5) and (6), we obtain

$$c_{\text{FSPO}}(L) \;\approx\; 2\,\Phi(-z),$$

which is independent of $L$ and hence preserves the length fairness required by Theorem 3.1. Moreover, as suggested by the Q–Q plot in Figure 1 (right), choosing $z < 2$ keeps us in a regime where the normal approximation is highly accurate.

We plot the theoretical clip–probability curves in Figure 2, together with the empirically observed clip fractions, showing close agreement with the theory.

## 5 METHOD: FSPO

For each prompt $x \sim \mathcal{D}$ we sample $G$ completions $\{y_i\}_{i=1}^{G} \sim \pi_{\theta_{\text{old}}}(\cdot \mid x)$ and optimize the PPO-style pessimistic surrogate

$$\mathcal{J}_{\text{FSPO}}(\theta) \;=\; \mathbb{E}_{x,\,\{y_i\}}\left[\frac{1}{G}\sum_{i=1}^{G}\min\Big\{\exp\big(S_\theta(y_i \mid x)\big)\,\widehat{A}_i,\; \exp\big(\text{clip}\big(S_\theta(y_i \mid x), -b_{L_i}, b_{L_i}\big)\big)\,\widehat{A}_i\Big\}\right], \tag{7}$$

where $\widehat{A}_i$ is an advantage estimate and $\text{clip}(s, \ell, u) = \min\{\max\{s, \ell\}, u\}$. The sequence-level log importance ratio is

$$S_\theta(y_i \mid x) \;=\; \log\frac{\pi_\theta(y_i \mid x)}{\pi_{\theta_{\text{old}}}(y_i \mid x)} \;=\; \sum_{t=1}^{L_i}\log\frac{\pi_\theta(y_{i,t} \mid h_{i,t})}{\pi_{\theta_{\text{old}}}(y_{i,t} \mid h_{i,t})}, \tag{8}$$

with $L_i = \text{len}(y_i)$ and $h_{i,t} = (x, y_{i,<t})$ the prefix at step $t$. FSPO performs *log-space* clipping by truncating $S_\theta$ to $[-b_L, b_L]$ before exponentiation, using the band as discussed in Section 4

$$b_L \;=\; \underbrace{\hat{\mu}_L}_{\text{drift}} \;+\; \underbrace{z\,\hat{\sigma}\,\sqrt{L}}_{\text{scale}}. \tag{9}$$

Note that in Equation (7) we average over the number of sequences $G$. This is natural for token-level clipping, but at sequence-level, clipping is applied to the entire sequence and the clip fraction is typically much larger. A natural idea would be to exclude the clipped sequences from averaging. However, we keep $G$ as the denominator, which serves as a dynamic step-size adjustment: when the clip fraction is higher which indicates that current mini-batch is unstable with higher variance, keeping the denominator at $G$ correspondingly yields a smaller effective update for that batch.

**Drift term.** Following Section 4, we set $\hat{\mu} = 0$ in our experimental settings. The drift connects to token-level KLs:

$$\mathbb{E}_{\pi_{\theta_{\text{old}}}}[S_L] \;=\; \mathbb{E}_{\pi_{\theta_{\text{old}}}}\left[\sum_{t=1}^{L}\log\frac{\pi_\theta(y_t \mid h_t)}{\pi_{\theta_{\text{old}}}(y_t \mid h_t)}\right] \;=\; \sum_{t=1}^{L} -D_{\text{KL}}\big(\pi_{\theta_{\text{old}}}(\cdot \mid h_t)\,\|\,\pi_\theta(\cdot \mid h_t)\big). \tag{10}$$

This further justifies our choice of $\hat{\mu} = 0$, as we empirically observe that the KL between the old and new policies is very small. However, this setting of $\hat{\mu} = 0$ is not required for FSPO. In fact, for substantially different regimes where the approximation $\mu \approx 0$ no longer holds (e.g., image generation (Wang et al., 2025a)), the insights FSPO provides about the sequence log-ratio distribution encourage practitioners to design drift treatments according to their own setting. Concretely, one can leverage empirical observations of $\hat{\mu}$ or adopt an adaptive scheme (e.g., a running-average estimate similar to our adaptive estimate for the scale term $\hat{\sigma}$ in section 7.5) to track $\hat{\mu}$ and adjust the clipping band $b_L$ accordingly, so that the clipping distribution remains well behaved.

**Scale term.** In our experimental setting, we use $c := z\,\hat{\sigma}$ as a single hyperparameter and set its value according to the observed estimate $\hat{\sigma} \approx 0.03$ according to pilot runs. In different domains, this fixed estimate may not transfer, and pilot runs or extensive hyperparameter tuning can be expensive.

Thus, we also propose dynamically estimating $\hat{\sigma}$ during training and adaptively adjusting $b_L$; we validate and discuss this variant in Section 7.5.

A natural extension is to use asymmetric scales $c_{\mathrm{upper}}$ and $c_{\mathrm{lower}}$, allowing separate control of the upper and lower clip ranges as in Yu et al. (2025). Current RL implementations commonly include *dual-clip* (Ye et al., 2020), which effectively clips the ratio at $(1 + \epsilon_{\mathrm{dual}})$ when $A < 0$; in our experiments, we also implement dual-clip in log-space and tune $c_{\mathrm{dual}}$. Implementation details are provided in Appendix C.2.

**Compatibility with other components** FSPO is a lightweight, plug-in modification that only changes the importance-ratio term in the policy loss. To isolate its effect, our implementation keeps all remaining components identical to the baselines (e.g., GRPO-style advantage). FSPO is compatible with alternative advantage estimators (e.g., $\widehat{A}^{\mathrm{LOO}}$ (Kool et al., 2019; Liu et al., 2025)), data filtering (Yu et al., 2025), and overlength penalties (Yu et al., 2025), among others.

# 6 EXPERIMENTAL SETUP

## 6.1 MODELS AND DATA

We evaluate our method on two base LLMs: **Qwen3-1.7B-Base** and **Qwen3-8B-Base** (Team, 2025). For training, we use DAPO-Math-17K (Yu et al., 2025) together with AIME problems up to and including 2023 (Mathematical Association of America, American Mathematics Competitions, 2024), accessed via (Veeraboina, 2024). Evaluation is conducted on held-out math benchmarks: MATH500 (Hendrycks et al., 2021), AIME24 (Maxwell-Jia, 2025), and AIME25 (OpenCompass, 2025). We exclude the MATH500 training split, as its difficulty is comparatively lower and does not significantly benefit training efficiency in our setting. We report **Avg@8** on MATH500 and **Avg@32** on AIME24/AIME25; here Avg@$k$ denotes per-instance accuracy averaged over $k$ independently sampled completions:

$$\mathrm{Avg}@k \;=\; \frac{1}{N} \sum_{i=1}^{N} \frac{1}{k} \sum_{j=1}^{k} a_{i,j}, \quad a_{i,j} \in \{0, 1\}.$$

Since each AIME set contains only 30 questions, using $k = 32$ yields more stable estimates. Detailed sampling configurations for evaluation are provided in Appendix C.

## 6.2 TRAINING FRAMEWORK

We build on VERL (Sheng et al., 2025) with vLLM (Kwon et al., 2023) as the rollout backend and Megatron-LM (Shoeybi et al., 2019) as the training backend. All models are trained under identical sampling configurations, batch sizes, and total token budgets. Full hyperparameters and infrastructure details are provided in Appendix C.

## 6.3 BASELINES

We compare against sequence-level RL baselines: **RLOO**, **GSPO**, and our **FSPO**. We also include **GRPO** to highlight the advantages of sequence-level importance sampling when properly designed. All methods share the same data, sampling configuration, batch size, and number of training steps; **FSPO** differs only in employing log-space clipping with a length-scaled band. For the **RLOO** baseline, we adopt the policy-loss formulation described in Section 2, but use the GRPO-style advantage $\hat{A}^{\mathrm{GRPO}}$ rather than $\hat{A}^{\mathrm{LOO}}$ for a fair comparison with the other three methods.

# 7 RESULTS AND ANALYSIS

## 7.1 MAIN RESULTS

Table 1 reports results on MATH500 (Avg@8) and AIME24/25 (Avg@32) for two base model sizes. For each method we show the *best* checkpoint (peak score across saved checkpoints) and

the *last* checkpoint (checkpoint of the last step). Overall, **FSPO** delivers consistent gains, with the largest margins on the harder AIME benchmarks and larger model size.

On **Qwen3-1.7B-Base**, **FSPO** attains the best AIME24 score (10.83/10.83) and the best last-average overall (29.16). On **Qwen3-8B-Base**, **FSPO** consistently outperforms the other methods across all benchmarks, achieving best/last averages of **49.79/48.98**, surpassing GRPO (+2.13/+1.93), RLOO (+1.99/+2.84), and GSPO (+2.82/+2.15). Gains are most pronounced on the more challenging AIME24 and AIME25: On AIME24, FSPO reaches **34.48/34.06**, yielding sizable gains versus GRPO (+3.23/+3.02), RLOO (+2.29/+4.06), and GSPO (+4.27/+3.85). On AIME25, FSPO achieves **24.69/24.69**, outperforming GRPO (+1.77/+2.19), RLOO (+1.67/+3.86), and GSPO (+2.19/+2.61).

Overall, gains of FSPO grow with model scale and task difficulty. This is expected as larger models and harder tasks induce broader, more heterogeneous response-length distributions, a regime where **FSPO**'s length-fair clipping yields the largest benefits.

| Method | MATH500 (Best/Last) | AIME24 (Best/Last) | AIME25 (Best/Last) | Average (Best/Last) |
|---|---|---|---|---|
| **Qwen3-1.7B-Base** | | | | |
| base | 52.20 | 3.02 | 3.33 | 19.52 |
| GRPO | 66.80/66.20 | 9.17/7.71 | 5.21/5.21 | 27.06/26.37 |
| RLOO | **70.80/70.80** | 10.73/7.60 | **6.77/6.77** | **29.43**/28.39 |
| GSPO | 69.00/69.00 | 9.48/9.48 | 6.04/6.04 | 28.17/28.17 |
| FSPO (ours) | 70.20/70.20 | **10.83/10.83** | 6.46/6.46 | 29.16/**29.16** |
| **Qwen3-8B-Base** | | | | |
| base | 71.20 | 10.00 | 10.00 | 30.40 |
| GRPO | 88.80/87.60 | 31.25/31.04 | 22.92/22.50 | 47.66/47.05 |
| RLOO | 88.20/87.60 | 32.19/30.00 | 23.02/20.83 | 47.80/46.14 |
| GSPO | 88.20/**88.20** | 30.21/30.21 | 22.50/22.08 | 46.97/46.83 |
| FSPO (ours) | **90.20/88.20** | **34.48/34.06** | **24.69/24.69** | **49.79/48.98** |

Table 1: Performance across benchmarks. "base" indicates the performance of the starting base model without RL training. MATH500 uses Avg@8; AIME24/AIME25 use Avg@32. Each cell shows **Best/Last** results. Bold indicates the best within each column.

## 7.2 LENGTH-FAIRNESS DIAGNOSTICS

We examine the clip fraction as a function of response length and compare the theoretical curve $c(L)$ predicted by equation 5 and 6; see Figure 2. The observed clip fractions match the theoretical patterns, where **RLOO** clips more frequently as length increases especially on short to medium lengths, **GSPO** shows a clear decreasing trend with length, and **FSPO** remains comparatively flat across lengths. Slightly higher values in the shortest-length bins in FSPO are due to limited samples and occasional outliers of abnormally short sequences.

The scale gap between the theoretical and empirical curves is expected due to the asymmetry between the upper and lower clip ranges in implementation and the skew of positive vs. negative advantages: PPO's pessimistic min surrogate effectively upper-bounds clipping for positive-advantage samples only (and vice versa).

For LRE, we compute the *acceptance* rate $q(L) = 1 - c(L)$ and exclude anomalously short cases with $L < 1000$. The resulting LREs are $0.162$ for RLOO, $0.264$ for GSPO, and $0.037$ for FSPO, where FSPO achieves the smallest LRE, according with its best performance demonstrated in 8B experiments.

## 7.3 EFFECTIVENESS: LEARNING DYNAMICS AND LENGTH STABILITY

As shown in Figure 3, both RLOO and FSPO learn quickly and increase response length early in training. However, RLOO's response length later explodes to very large values, with much of the additional content being filler. A plausible explanation is that longer sequences are more likely to be clipped under RLOO; consequently, negative signals from long incorrect answers are suppressed, and the model fails to regulate length. Moreover, as responses grow longer, RLOO's higher clip

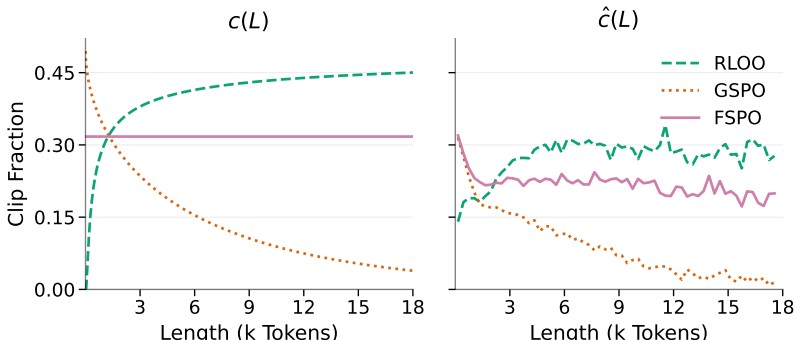

Figure 2: **Theoretical and empirical clip fraction. Left:** Theoretical clip probability $c(L)$ computed from Equations (5) and (6) using the hyperparameters in Appendix C.2, where we set $\xi = \log(1 + c_{\text{upper}})$. **Right:** Observed clip fraction $\hat{c}(L)$ with bin size $= 200$, collected from the experiments on Qwen3-8B-Base model.

probability hampers learning and reward improvements plateau, whereas FSPO continues to make steady gains. By contrast, GSPO learns more slowly at the beginning and struggles to increase length, especially for the 1.7B model. On the 8B model, GSPO attains high rewards near or comparable to FSPO during training, yet its evaluation performance is suboptimal, indicating that length imbalance during training can impair calibration during generalization evaluation. FSPO attains the best performance with moderate average length on the 8B model, suggesting more balanced learning across lengths and more effective use of length.

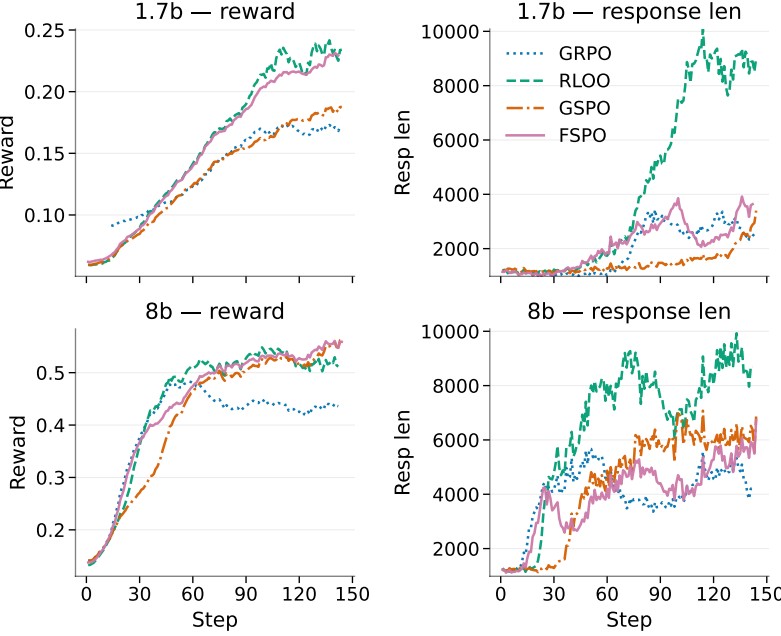

Figure 3: **Learning dynamics during training.** Left column: mean reward (1.7B and 8B). Right column: mean response length (1.7B and 8B). Reward curves are smoothed with EMA for visualization.

To further assess downstream behavior, we report the *overlong rate* (the proportion of samples that reach the maximum response length and are truncated) and mean response length after excluding overlong samples. FSPO exhibits a markedly lower overlong rate, indicating stable control of response length. In contrast, methods with incorrect importance weights (GRPO, GSPO) show substantially higher overlong rates, even thoguh their mean lengths after excluding overlong samples

are similar, leading to suboptimal behavior. RLOO displays both higher overlong rate and larger length. Detailed statistical analysis can be found in Appendix D.

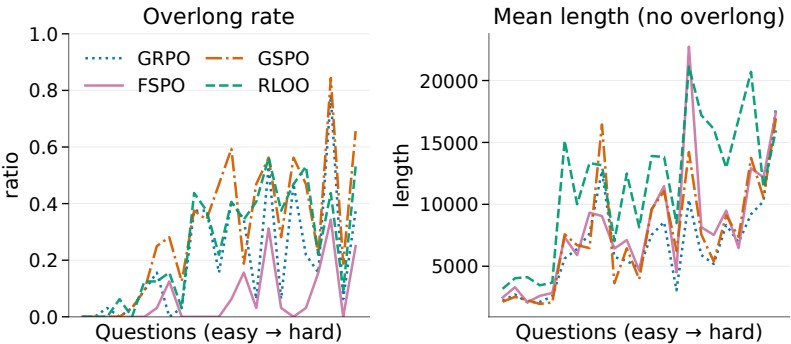

Figure 4: **Overlong rate and mean response length** on AIME24. We plot evaluation-time sampling; the x-axis orders the 30 problems from easy to hard, where difficulty is measured by the overall average accuracy across the four methods.

## 7.4 ABLATION STUDY: LARGER CLIP RANGE

As in Figure 2, RLOO's clip fraction is large, potentially due to its relatively small clip range (Appendix C.2). Note that in FSPO the *ratio-level* clip range for a sequence with $L = 10{,}000$ is $\exp(\sqrt{10000} \times 0.03) = 20.09$, much larger than the $1.667$ $(1 + c_{\text{upper}})$ used in RLOO. Thus, one may hypothesize that FSPO's gains stem from being more permissive on long sequences than RLOO. To disentangle this, we evaluate RLOO with a *fixed* larger clip range (upper $= 20$, lower $= 0.95$). As shown in Figure 5, this variant does not improve performance and can even be worse than standard RLOO. This indicates that *length fairness*, rather than mere leniency toward long responses, is key to FSPO's effectiveness.

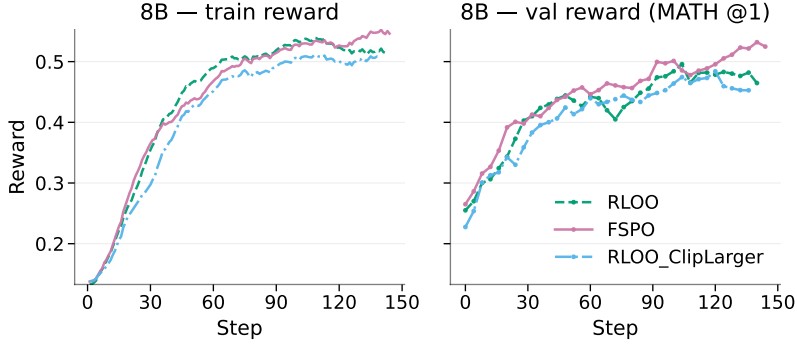

Figure 5: **Ablation: fixed larger clip range. Left:** mean rewards during training. **Right:** validation curves during training.

## 7.5 FSPO WITH ADAPTIVE SCALE TERM

In this section, we implement FSPO with an adaptive scale term as discussed in Section 5 and evaluate it on Qwen3-1.7B-Base. We study three possible choices for the adaptive estimator:

- **Per-batch standard deviation.** $\hat{\sigma}^{(i)}$ computes the observed standard deviation for the $i$-th batch.
- **Cumulative Moving Average (CMA).** $\hat{\sigma}^{(i+1)}_{\text{CMA}} = \frac{i}{i+1}\hat{\sigma}^{(i)}_{\text{CMA}} + \frac{1}{i+1}\hat{\sigma}^{(i)}$.
- **Exponential Moving Average (EMA).** $\hat{\sigma}^{(i+1)}_{\text{EMA}} = (1 - \alpha)\hat{\sigma}^{(i)}_{\text{EMA}} + \alpha\hat{\sigma}^{(i)}$. $\alpha = 0.1$

The plot for the three estimates is shown in Appendix C.4. We select $\hat{\sigma}_{\text{EMA}}$ as the adaptive estimator, as it yields fewer fluctuations yet is also less affected by large values in the early phase of training. Specifically, we compute $\hat{\sigma}_{\text{EMA}}^{(i)}$ at step $i$ and adapt the clipping range $b_L$ for the next step accordingly. We denote FSPO with fixed scale term as **FSPO_fix**, and the adaptive as **FSPO_ada**. Evaluation results are shown in Table 2, and the training reward and validation curves are provided in Figure 6 as complementary evidence. From the reward curves, we observe that in the early stage of training, FSPO_ada uses a larger clipping range due to the higher variance of the sequence log ratios, which introduces some instability and leads to slightly slower reward growth compared to FSPO_fix. However, in the middle and late stages of training, FSPO_ada catches up with and even surpasses FSPO_fix in terms of rewards, and the final performance is competitive or slightly better.

| Method | MATH500 (Best/Last) | AIME24 (Best/Last) | AIME25 (Best/Last) | Average (Best/Last) |
|---|---|---|---|---|
| **Qwen3-1.7B-Base** | | | | |
| GRPO | 66.80/66.20 | 9.17/7.71 | 5.21/5.21 | 27.06/26.37 |
| FSPO_fix | 70.20/**70.20** | **10.83/10.83** | 6.46/6.46 | 29.16/**29.16** |
| FSPO_ada | **70.40**/70.20 | 10.64/10.64 | **6.53/6.53** | **29.19**/29.12 |

Table 2: Performance comparison between FSPO with fixed and adaptive scale terms across benchmarks. Evaluation methods are the same as in Table 1.

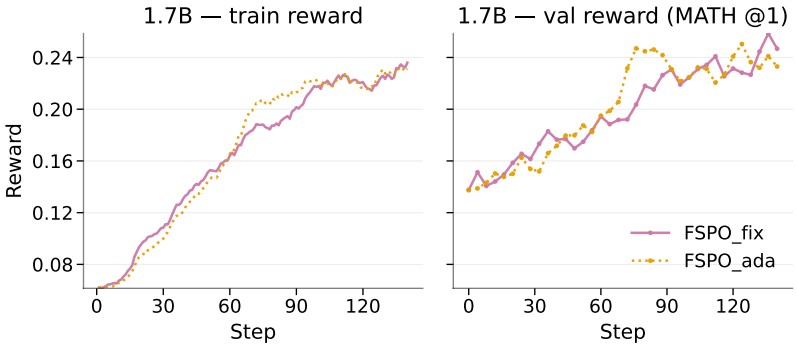

Figure 6: **Comparison between FSPO with fixed and adaptive scale terms on the 1.7B model. Left:** mean rewards during training. **Right:** validation curves.

## 8 CONCLUSION

We studied the clipping mechanism in sequence-level importance sampling (IS) for RLVR scenarios, showing that a fixed clip range induces a length–reweighting pathology that biases acceptance across response lengths and distorts the effective objective. We formalized *length fairness* via the Length Reweighting Error (LRE) and established a cosine–direction guarantee linking small LRE to update–direction fidelity. Guided by an approximate Gaussian law for the sequence log–IS sum, we proposed **FSPO**: clipping in log–IS space with a $\sqrt{L}$-scaled band that preserves IS semantics while equalizing acceptance across lengths. Empirically, on three math benchmarks and two model scales, FSPO flattens acceptance–by–length and delivers consistent gains, with the largest improvements on the 8B model. We also develop and validate an adaptive-scale variant of FSPO that tracks the log–IS variance online, avoiding hand-tuning clipping range for different task domains. Overall, FSPO is a simple, intuitive, and practical algorithmic modification with enhanced performance and promising transferability. Looking ahead, we plan to extend evaluation to broader RLVR settings and combine FSPO with stronger advantage estimation, aiming to build more capable RL pipelines.

## REPRODUCIBILITY STATEMENT

We make a concerted effort to ensure the reproducibility of our work. We describe the algorithm and implementation notes in detail in Section 5 and the algorithmic hyperparameters in Appendix C.2. We provide the full experimental setup in Section 6, and we give a detailed description of the infrastructure, framework, training configuration, and evaluation configuration in Appendix C, where we also include the settings for all of our baseline method experiments.

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

## A  WHY SEQUENCE-LEVEL IS FOR RLVR

RLOO (Ahmadian et al., 2024) models the entire generation as a single action (a bandit setting), but its context is RLHF and it does not fully analyze the inadequacy of token-level IS and the correctness of sequence-level IS for RLVR. GSPO (Zheng et al., 2025) notes that, in RLVR, the granularity of importance sampling should match that of the reward, but does not provide a detailed justification. Here we offer a more explicit discussion.

As shown by Kakade & Langford (2002); Schulman et al. (2015a) (see also Levine (2019)), the improvement of the objective between old and new parameters can be written as

$$J(\boldsymbol{\theta}) - J(\boldsymbol{\theta}_{\text{old}}) = \mathbb{E}_{\tau \sim \pi_{\boldsymbol{\theta}}}\left[\sum_{t=0}^{\infty} \gamma^t A^{\pi_{\boldsymbol{\theta}_{\text{old}}}}(s_t, a_t)\right] \tag{11}$$

$$= \sum_{t=0}^{\infty} \mathbb{E}_{s_t \sim p_{\boldsymbol{\theta}}(s_t)}\left[\mathbb{E}_{a_t \sim \pi_{\boldsymbol{\theta}}(a_t|s_t)}\left[\gamma^t A^{\pi_{\boldsymbol{\theta}_{\text{old}}}}(s_t, a_t)\right]\right] \tag{12}$$

$$= \sum_{t=0}^{\infty} \mathbb{E}_{s_t \sim p_{\boldsymbol{\theta}}(s_t)}\left[\mathbb{E}_{a_t \sim \pi_{\boldsymbol{\theta}_{\text{old}}}(a_t|s_t)}\left[\frac{\pi_{\boldsymbol{\theta}}(a_t|s_t)}{\pi_{\boldsymbol{\theta}_{\text{old}}}(a_t|s_t)} \gamma^t A^{\pi_{\boldsymbol{\theta}_{\text{old}}}}(s_t, a_t)\right]\right], \tag{13}$$

where $p_{\boldsymbol{\theta}}(s_t)$ is the $\gamma$-discounted state visitation under $\pi_{\boldsymbol{\theta}}$. This is where token-level importance sampling naturally appears: we must express expectations under $\pi_{\boldsymbol{\theta}}$ using samples drawn from $\pi_{\boldsymbol{\theta}_{\text{old}}}$. Note that the *state* distribution $p_{\boldsymbol{\theta}}(s_t)$ is not corrected by IS (it factors through all previous actions, and naively correcting it leads to high variance). TRPO's trust region and PPO's clipping are introduced precisely to control the mismatch between $p_{\boldsymbol{\theta}_{\text{old}}}(s_t)$ and $p_{\boldsymbol{\theta}}(s_t)$ when policies are close.

However, this formulation becomes problematic in the RLVR setting, where all tokens of a sequence share a single sequence-level advantage. From equation 11 to equation 12, the expectations over $(s_{t+1}, a_{t+1}, s_{t+2}, \ldots)$ are marginalized out; that step requires the summand to depend only on $(s_t, a_t)$ (not on the *future* of the trajectory). This condition fails in RLVR, in which $A(s_t, a_t) = A(\tau)$ for all $t$ in a sampled sequence.

To make this concrete, consider two completions $\boldsymbol{y}_a, \boldsymbol{y}_b \sim \pi_{\boldsymbol{\theta}_{\text{old}}}(\cdot \mid \boldsymbol{x})$ that share a prefix up to index $t$:
$$\boldsymbol{y}_a = (y_0, y_1, \ldots, y_t, y_{t+1}^{(a)}, y_{t+2}^{(a)}, \ldots), \qquad \boldsymbol{y}_b = (y_0, y_1, \ldots, y_t, y_{t+1}^{(b)}, y_{t+2}^{(b)}, \ldots).$$
Suppose $\boldsymbol{y}_a$ is correct while $\boldsymbol{y}_b$ is incorrect (e.g., $A(\boldsymbol{y}_a) = 0.5$, $A(\boldsymbol{y}_b) = -0.5$), with $\pi_{\boldsymbol{\theta}_{\text{old}}}(\boldsymbol{y}_a) = \pi_{\boldsymbol{\theta}_{\text{old}}}(\boldsymbol{y}_b)$ and $\pi_{\boldsymbol{\theta}}(\boldsymbol{y}_a) > \pi_{\boldsymbol{\theta}}(\boldsymbol{y}_b)$. Thus, conditioned on the shared prefix $(y_0, \ldots, y_t)$, the current policy $\pi_{\boldsymbol{\theta}}$ makes the correct continuation more likely than the incorrect one. Intuitively, the next update should *upweight* the shared-prefix gradients. Sequence-level IS achieves this because

$$1\frac{\pi_{\boldsymbol{\theta}}(\boldsymbol{y}_a)}{\pi_{\boldsymbol{\theta}_{\text{old}}}(\boldsymbol{y}_a)} \;>\; \frac{\pi_{\boldsymbol{\theta}}(\boldsymbol{y}_b)}{\pi_{\boldsymbol{\theta}_{\text{old}}}(\boldsymbol{y}_b)},$$

so the shared-prefix tokens receive larger weights for $\boldsymbol{y}_a$ than for $\boldsymbol{y}_b$. By contrast, token-level IS assigns the same per-token ratios to the shared tokens $(y_0, \ldots, y_t)$ for both sequences (and for any other sample with that prefix), so it cannot express this desirable preference. This illustrates why sequence-level IS is the right granularity for RLVR.

Interestingly, with sequence-level IS the $p_{\boldsymbol{\theta}_{\text{old}}}$ vs. $p_{\boldsymbol{\theta}}$ state-visitation mismatch that motivates TRPO/PPO is mitigated at the sequence level: the *entire* trajectory probability is corrected by the sequence ratio. Nevertheless, clipping remains necessary to control the variance of the sequence ratio, hence our focus on well-designed sequence-level clipping.

## B  PROOF OF THEOREM 3.1

**Cosine lemma.**  For nonzero vectors $\boldsymbol{u}, \boldsymbol{v}$, $\cos \angle(\boldsymbol{u}, \boldsymbol{v}) \geq \frac{\|\boldsymbol{v}\| - \|\boldsymbol{u} - \boldsymbol{v}\|}{\|\boldsymbol{v}\| + \|\boldsymbol{u} - \boldsymbol{v}\|}$.

**Proof.**  Note that $\cos \angle(\boldsymbol{g}^b, \boldsymbol{g}^\star) = \cos \angle(\boldsymbol{g}^b, \bar{q}\boldsymbol{g}^\star)$. To apply the cosine lemma, we want to bound $\|\boldsymbol{g}^b - \bar{q}\boldsymbol{g}^\star\|$. Recall $\boldsymbol{g}^b = \mathbb{E}_L[q(L)\boldsymbol{g}_L^b]$, $\boldsymbol{g}^\star = \mathbb{E}_L[\boldsymbol{g}_L^\star]$, and $\bar{q} = \mathbb{E}[q(L)]$. Decompose
$$\boldsymbol{g}^b - \bar{q}\boldsymbol{g}^\star = \mathbb{E}_L\big[(q(L) - \bar{q})\boldsymbol{g}_L^\star\big] \;+\; \mathbb{E}_L\big[q(L)(\boldsymbol{g}_L^b - \boldsymbol{g}_L^\star)\big].$$

By the triangle inequality,

$$\left\| \boldsymbol{g}^{\,b} - \bar{q}\,\boldsymbol{g}^\star \right\| \leq \underbrace{\mathbb{E}_L\big[\, |q(L) - \bar{q}|\, \|\boldsymbol{g}_L^\star\| \,\big]}_{\text{cross-length reweighting}} + \underbrace{\mathbb{E}_L\big[\, \|\, q(L)\,(\boldsymbol{g}_L^{\,b} - \boldsymbol{g}_L^\star)\| \,\big]}_{\text{within-length stratification}}$$

$$\leq \bar{q}\,\mathbb{E}_L\left[\left|\frac{q(L)}{\bar{q}} - 1\right| \|\boldsymbol{g}_L^*\|\right] + \eta\,\mathbb{E}_L[\|(q(L) - \bar{q})\,\boldsymbol{g}_L^* + \bar{q}\,\boldsymbol{g}_L^*\|] \quad \text{(by Assumption 3.1)}$$

$$\leq (1 + \eta)\,\bar{q}\,\mathbb{E}_L\left[\left|\frac{q(L)}{\bar{q}} - 1\right| \|\boldsymbol{g}_L^*\|\right] + \eta\,\bar{q}\,\mathbb{E}_L[\|\boldsymbol{g}_L^*\|]$$

$$\leq \bar{q}\,\big(2\gamma(1 + \eta)\,\mathrm{LRE} + \eta\big)\,\mathbb{E}_L[\|\boldsymbol{g}_L^*\|] \quad \text{(by Assumption 3.2 and the definition of LRE).}$$

Since $\mathbb{E}_L[\|\boldsymbol{g}_L^*\|] = \kappa\,\|\boldsymbol{g}^*\|$, applying the cosine lemma with $\boldsymbol{u} = \boldsymbol{g}^{\,b}$, $\boldsymbol{v} = \bar{q}\,\boldsymbol{g}^*$ yields Theorem 3.1.

**Weighted-LRE variant.** If one prefers to avoid the bounded co-variation assumption, define

$$\mathrm{LRE}_w = \frac{1}{2}\,\mathbb{E}\left[\left|\frac{q(L)}{\bar{q}} - 1\right| \frac{\|\boldsymbol{g}_L^\star\|}{\mathbb{E}[\|\boldsymbol{g}_L^\star\|]}\right].$$

The same argument gives the bound with LRE replaced by $\mathrm{LRE}_w$.

**More discussion: beyond length.** The proof of Theorem 3.1 does not rely on using $L$ specifically as the partitioning variable. It only requires that the sample space can be partitioned into groups where (i) cross-group signal magnitudes exhibit dispersion, and (ii) within-group stratification errors are controlled. Thus, $L$ can be replaced by, e.g., length bins (which justifies our diagnostic binning) or other structural attributes. A general design principle follows: a clipping mechanism should avoid introducing systematic bias across reasonable partitions, unless such bias is intentionally desired.

## C  CONFIGURATIONS

### C.1  TRAINING CONFIGURATIONS

We run all experiments on a single node with $8 \times \mathrm{H200}\,(140\,\mathrm{GB})$ GPUs. We report the configuration for 8B experiments here. Under the configuration below, one epoch takes approximately 3–4 days; the wall-clock time increases with the average response length.

Table 3: Training configuration.

| Item | Value |
|---|---|
| Prompt / response max | 2,000 / 18,000 tokens |
| Global batch size (sequences) | 128 |
| Minibatch size | 32 |
| Per-GPU microbatch | 32 |
| Total steps | 144 |
| Optimizer & LR | AdamW, $1 \times 10^{-6}$ |
| Parallelism | Megatron TP$= 8$ |
| Rollout $n$ | 16 |
| vLLM GPU util. | 0.5 |
| Seeds | 42 |

The training for 1.7B models shares similar configurations except for Megatron TP$= 2$ and vLLM GPU util.$= 0.7$. One epoch for 1.7B training takes less than or approximate to 1 day.

### C.2  ALGORITHMIC HYPERPARAMETERS AND TUNING

**Limitations.** Due to compute constraints, we did not perform an exhaustive hyperparameter search. For settings similar to ours, we recommend fixing the base clipping scale $c$ at $0.03$ or higher; this value may not transfer across substantially different datasets or model sizes.

Table 4: Algorithmic hyperparameters.

| Hyperparameter | Value |
|---|---|
| Upper clip $c_{\text{upper}}$ | 0.03 |
| Lower clip $c_{\text{lower}}$ | 0.03 |
| Dual clip $c_{\text{dual}}$ | 0.03 |
| use_KL | disabled |
| Entropy coefficient | 0 |
| Advantage estimator | GRPO-style |

Table 5: Baseline clipping configuration.

| Baseline | $c_{\text{upper}}$ | $c_{\text{lower}}$ | $c_{\text{dual}}$ |
|---|---|---|---|
| GRPO | 0.28 | 0.20 | 3.0 |
| RLOO | 0.667 | 0.40 | 3.0 |
| GSPO | $4 \times 10^{-4}$ | $3 \times 10^{-4}$ | disabled |

**Interpreting $c$, $z$, and $\hat{\sigma}$.** In FSPO, the sequence log-IS ratio $S$ is clipped with a symmetric length-dependent band whose *half-width* is

$$b_L = z\,\hat{\sigma}\,\sqrt{L} \equiv c\sqrt{L}, \qquad c := z\,\hat{\sigma}.$$

In our experiments we first obtain $\hat{\sigma}$ from a short baseline run and then fix $c$ (e.g., $c = 0.03$). To avoid a dedicated pilot run while preserving stability, a practical recipe is:

1. Initialize $z \in [1, 1.5]$ and a heuristic $\hat{\sigma}_0 = 0.03$ for warm-up.
2. Track a running estimate $\hat{\sigma}_t$ via EMA: $\hat{\sigma}_t \leftarrow (1 - \alpha)\hat{\sigma}_{t-1} + \alpha\,\text{std}_{\text{batch}}(S)$.
3. Update the clip band as $b_L = z\,\hat{\sigma}_t\,\sqrt{L}$ throughout training.

**Baseline hyperparameters.** We also report the clipping settings used for baselines. Note that these clip ranges are specified in the *ratio* space (conventional for PPO-style objectives), whereas FSPO clips in the *log* ratio with a $\sqrt{L}$ band. We adopt clip-higher for GRPO (Yu et al., 2025) and follow GSPO's guidance for its settings. The cliprange_c parameter controls dual-clip in VERL.

## C.3 TEST-TIME CONFIGURATIONS

We use OpenCompass (Contributors, 2023) as our evaluation framework.

Table 6: Decoding configuration.

| Item | Value |
|---|---|
| Temperature | 0.6 |
| Top-$p$ | 0.95 |
| Top-$k$ | 200 |
| Max generation tokens | 32,000 |
| Batch size | 256 |
| Tensor parallel | 8 |
| Data parallel | 1 |

## C.4 VARIANTS OF ADAPTIVE ESTIMATE FOR $\hat{\sigma}$

We plot the computed variants of estimate for $\hat{\sigma}$ in Figure 7.

## D STATISTICAL ANALYSIS FOR SECTION 7.3

To further substantiate the findings presented in Section 7.3, we provide supplementary statistical analyses in this section.

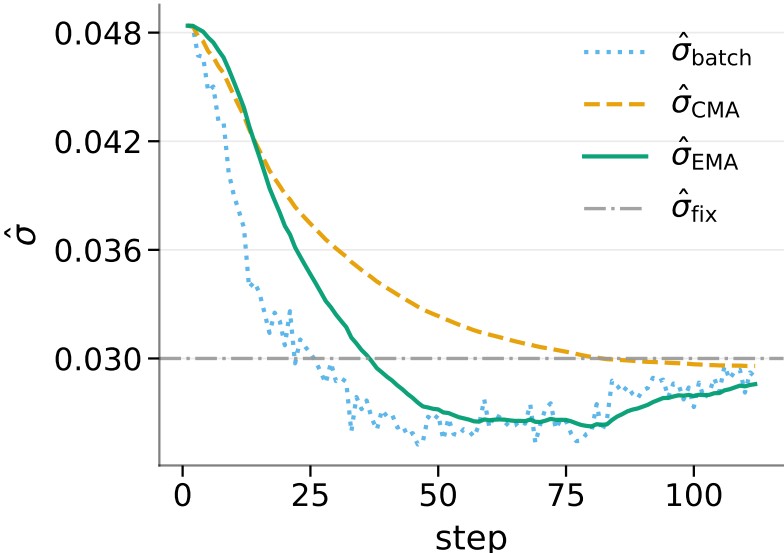

Figure 7: Variants of adaptively estimating the scale term $\hat{\sigma}$.

Regarding the overlong rate, we conduct permutation tests to verify the null hypothesis that FSPO exhibits an overlong rate identical to that of the other three variants. We perform these tests pairwise between FSPO and each baseline variant for every question, plotting the resulting $p$-values in Figure 8. The number of permutations is set to $n_{\text{perm}} = 100,000$.

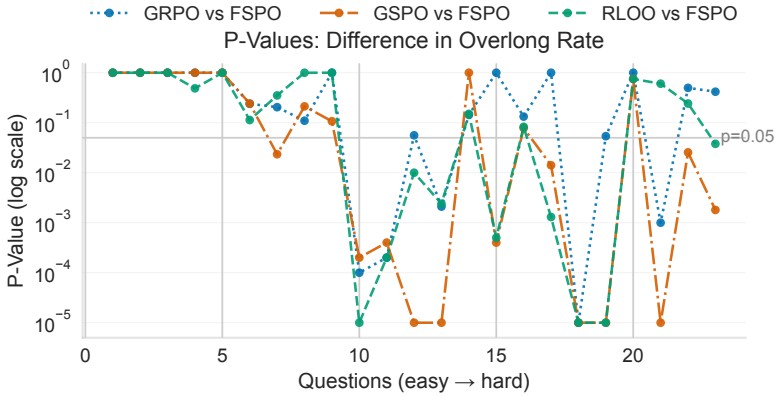

Figure 8: Permutation test $p$-values comparing the overlong rate of FSPO against other baselines across questions.

The results indicate that among the 14 harder questions (indexed from 10 to 23; the remaining 7 questions are excluded due to near-zero accuracy), FSPO demonstrates significantly difference ($p < 0.05$) in 64.29% (9/14) of cases compared to GRPO, 71.43% (10/14) compared to RLOO, and 85.71% (12/14) compared to GSPO, with FSPO consistently showing lower values.

For the mean response length (excluding overlong samples), we present the trends with confidence intervals in Figure 9 and the corresponding permutation test $p$-values in Figure 10. Indeed, we observe that RLOO yields a significantly higher mean length. However, for GRPO and GSPO, the results suggest a length distribution comparable to FSPO, with no significant differences observed in most cases.

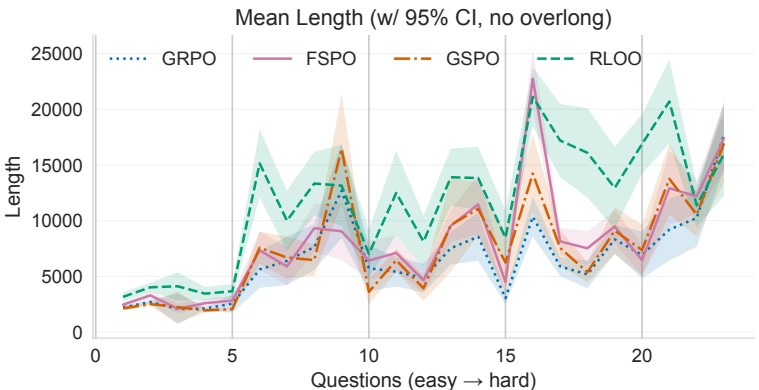

Figure 9: Mean response lengths with 95% CI.

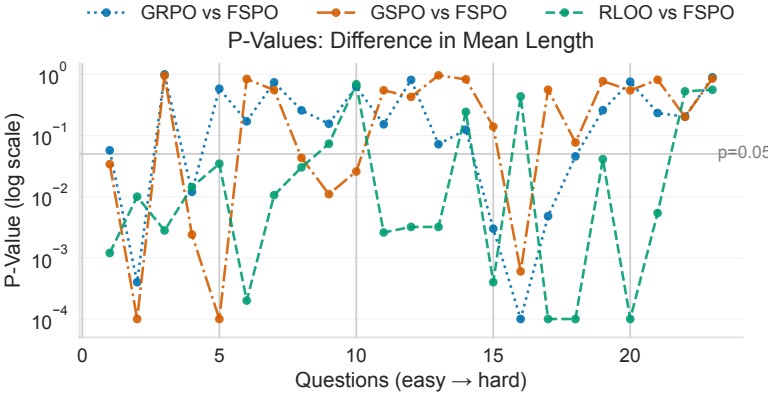

Figure 10: P-values of permutation test of mean lengths for each question between FSPO and other baselines.

We include the detailed list of "easy to hard questions" for AIME24 here. The numbers $> 15$ are the questions in the II problem set. Only 23 questions are listed; the remaining 7 questions are discarded for near 0 accuracies.

0, 18, 1, 3, 6, 14, 17, 4, 27, 20, 16, 15, 24, 21, 5, 9, 8, 2, 26, 19, 12, 25, 13

# E   USE OF LARGE LANGUAGE MODELS

We employed large language models (LLMs) in three ways:

**Language polishing.**   We used GPT-5 (OpenAI, 2025) to refine the writing of the abstract, Sections 1, 2 and 5 to 7 and the Appendix. Edits included suggesting idiomatic phrasing, improving clarity and style, and correcting grammar errors and typos.

**Literature search.**   Leveraging recent LLM browsing capabilities, we used GPT-5 to surface part of the relevant related work referenced in Section 2, and to compare evaluation frameworks in Section 6, which informed our choice of OpenCompass.

**Figure and table refinement.**   We used LLMs to improve table formatting and figure aesthetics, including layout suggestions, color palettes, and ensuring consistent visual style across plots.

