# OpenReview forum: "Clip Your Sequences Fairly: Enforcing Length Fairness for Sequence‑Level RL"
_ICLR.cc/2026/Conference — Submitted to ICLR 2026_

### Official Review · Reviewer_rLLd · 2025-10-28

**Soundness:** 3
**Presentation:** 3
**Contribution:** 3
**Rating:** 6
**Confidence:** 3

**Summary:**

This paper introduces a sequence-level reinforcement RL method designed for Large Language Models. The authors identify a mismatch in current methods (like GRPO and RLOO) that apply PPO-style fixed-range clipping to sequence-level importance-sampling rations. They argue that longer responses are clipped more frequently distorting the optimization direction. FSPO addresses this by clipping the sequence log-IS ratio using a dynamic band that scales with length.  FSPO is evaluated on math reasoning tasks using Qwen3 base models (1.7B and 8B), showing improved stability, flatter acceptance rates across length bins, and increased performance compared to the baselines.

**Strengths:**

FSPO preserves IS semantics while restoring fairness via the length scaled band. Another strenght is the theoretical contribution - the paper formalizes the problem via lenght reweighting error and provides a theorem linking this to update direction fidelity. It grounds its solution in the asymptotic Gaussian law of sequence log-IS ratios. The method shows improvement over strong baselines across multiple benchmarks and model scales, with the most significant improvements on harder tasks (AIME24/25) and larger models (8B)

**Weaknesses:**

The evaluations domain is a bit limited as it only considers math reasoning tasks. And while there are some gains in performance they are not huge. However the principled approach is nice.  Another limitation might be the drift assumption as this could change in different experimental settings.

**Questions:**

how do you expect the drift assumption to hold in significantly different experiment settings?

---

> ### Author Response · Authors · 2025-11-21
>
> Thank you for your thoughtful feedback!
>
> **Update of paper:**
>
> We have updated our paper to include an Adaptive FSPO variant (Section 7.5), which dynamically estimates the clipping scale using a running average estimator, eliminating the need for pilot runs for hyperparameter tuning while preserving performance.
>
> **Response to Weakness:**
>
> We appreciate the reviewer’s positive assessment of the principled nature of our approach. It is true that our evaluations are limited to math reasoning tasks. We chose this domain because, within RLVR, math is both central and challenging, and it naturally exhibits diversity in content and response length. In this setting, FSPO’s contribution is not only the performance improvement but also the more effective use and stabilization of response length.
> At the same time, FSPO is simple, intuitive, and easy to implement, and it provides actionable guidance for tuning when transferring to new tasks. As in our main experiments, one can estimate $\hat{\sigma}$ via a pilot run to set the scale $c$, or use the adaptive variant in Section 7.5 to estimate the clip range online. In both cases, the clip band adapts to the task-specific distribution, which we expect to extend naturally to other domains. Broader empirical validations on other RLVR tasks remains important future work.
>
> **Response to Question:**
>
> We appreciate this insightful observation. In fact, we do not expect the drift assumption to hold in other domains. The empirical choice $\hat{\mu} \approx 0$ is just a convenient choice and is specific to our current experimental setting where the estimated drift is small relative to $\hat{\sigma}$. We emphasize, however, that FSPO does not fundamentally require $\mu = 0$. We expect FSPO to remain applicable and potentially even more beneficial in regimes where $\mu \neq 0$.
>
> Interestingly, a very recent work GRPO-guard (https://www.arxiv.org/abs/2510.22319, we have added discussion and reference of it in the Method section in our updated paper) applying GRPO to **flow-matching** for image generation reports a systematic bias of the sequence-level IS ratio below 1 (which corresponds to $\mu < 0$ for our log-IS). That work observes that this drift induces asymmetric clipping: GRPO’s clipping effectively becomes looser for positive-weight samples, leading to over-optimization, and they propose normalizing the IS weights as a remedy.
>
> FSPO’s perspective on the distribution of sequence log-IS ratios is actually well suited for this situation: it explicitly encourages practitioners to observe the empirical distribution of the log-IS ratio and to adjust the clip band $b_L$ dynamically (e.g., via a pilot run or an adaptive running estimate similar to $\hat\sigma$) so that clipping remains symmetric and well-calibrated. This can address the asymmetry issue while preserving the IS semantics (unlike directly normalizing the ratios). Note that symmetric clipping is not the original primary motivation of FSPO: our main goal is to balance clip fractions across lengths, but in the presence of nonzero drift it becomes a useful side effect that fixed clip ranges in baseline methods cannot provide.

---

### Official Review · Reviewer_wUha · 2025-10-30

**Soundness:** 3
**Presentation:** 3
**Contribution:** 3
**Rating:** 6
**Confidence:** 3

**Summary:**

This paper introduces FSPO which  propose a length-scaled clipping band in log-IS space to equalize acceptance rates across sequence lengths. Empirical results on math benchmarks (MATH500, AIME24/25) show FSPO outperforms existing methods.

**Strengths:**

1. this paper address a critical issue in RLVR.
2. method is intuitive and simple to implement.

**Weaknesses:**

1. Author did not thoroughly study the clipping hyperparameters and their effect. This is crucial because the choice of clipping band can significantly influence variance reduction and fairness.

2. Some assumptions may be too strict. For assumption 3.1, clipping may affect sequences with high variance and especially when the data is limited.  In such cases, clipping might distort the distribution and negatively impact learning.

**Questions:**

1. As Figure 2 "The scale gap between the theoretical and empirical curves is expected due to the asymmetry between the upper and lower clip ranges in implementation." Can adaptively adjust the clip band further improve performance e.g. reduce the gap? An adaptive strategy might better handle varying sequence lengths and variance.

2. Figure 4 has very large variance which makes hard to make reasonable conclusion, can author consider plot it with confidence interval or std? This would help clarify the statistical significance of the observed patterns and better support the claims regarding length fairness.

3. In Table 3, author list upper clip, lower clip and dual clip. Can author give more explanation on how this dual clip affect the fairness or stability?

---

> ### Author Response · Authors · 2025-11-21
>
> Thank you for your thoughtful feedback!
>
> **Update of paper:**
>
> We have updated our paper to include an Adaptive FSPO variant (Section 7.5), which dynamically estimates the clipping scale using a running average estimator, eliminating the need for pilot runs for hyperparameter tuning while preserving performance.
>
> **Response to Weakness 1:**
>
> For the first concern, we indeed did not perform extensive hyperparameter tuning due to compute limitations. However, we would like to emphasize that FSPO still offers better guidance for transferability than existing RLVR baselines. Methods such as GRPO and DAPO expose only fixed heuristic clip ranges (e.g., 0.28 / 0.20) and reuse these values across different settings without principled instructions for retuning. In contrast, FSPO sets the base clipping scale (c) from the empirically observed per-length standard deviation $\hat{\sigma}$ of the sequence log-IS ratio. Although estimating $\hat{\sigma}$ requires pilot run, this procedure provides explicit guidance on how to choose (c) when transferring to new domains.
>
> Moreover, to further reduce the cost of such pilot runs, in the updated version we introduce an adaptive variant that estimates $\hat{\sigma}$ using a running average estimator and adjusts the clip range online. This adaptive FSPO is implemented and empirically validated in Section 7.5, and it further reduces the hyperparameter-tuning overhead while preserving performance, showing FSPO's potential for adaptively maintaining clip fairness in different distributions.
>
> **Response to Question 1:**
>
> The “asymmetry” mentioned in Figure 2 refers to our use of PPO-style pessimistic clipping: we clip the upper side only for positive-advantage samples and the lower side only for negative-advantage samples. In practice, the counts of positive and negative advantage samples are not symmetric, so the empirical clip fractions deviate from the idealized symmetric clip probabilities used in the theoretical curves (as well as noise).
> As discussed in our response to Weakness 1, we also add an adaptive strategy that estimates $\hat{\sigma}$ online and updates the band. This adaptive variant, evaluated in Section 7.5, handles varying sequence lengths and variance more flexibly and achieves competitive performance to the fixed-band FSPO.
>
> **Response to Weakness 2:**
>
> The high-level goal of clipping in practice is to remove a small number of extreme samples so as to reduce variance and avoid unstable gradient estimates, especially when the sampled data are limited. Assumption 3.1 formalizes the requirement that, considering the whole sample space, the average gradient inside the unclipped region is close to the true full gradient **in expectation**; i.e., clipping should not introduce substantial systematic bias on its own. We agree that this requires some assumptions on the underlying distribution, but we also believe that reasonable clipping mechanisms should satisfy this property, or one would need to enlarge the clip band accordingly.
>
> **Response to Question 2:**
>
> This is a very helpful suggestion! In the updated version, we add rigorous statistical analysis in Appendix D. Specifically, for the overlong rate, we apply a permutation test and show that FSPO achieves significantly lower overlong rates on most of the hard problems, showing enhanced stability. For the mean length, we plot curves with confidence intervals and again apply permutation tests. The results indicate that, in a statistically significant sense, we can only conclude that FSPO’s mean length is smaller than RLOO’s, while the mean lengths of GSPO and GRPO are similar to FSPO’s. However, this does not affect our main conclusions.
>
> **Response to Question 3:**
>
> The “upper clip” and “lower clip” in Table 3 refer to the standard PPO-style clipping: the upper bound is applied only to positive-advantage samples, and the lower bound only to negative-advantage samples. We additionally include a “dual clip” parameter because this is the default setting in the VERL implementation for many RL algorithms; practically, dual clip is the upper bound applied to negative-advantage samples. In sequence-level RL with a fixed clip range, this dual-clip mechanism also introduces length-dependent bias. To keep dual clip length-fair, we apply the same $\sqrt{L}$-scaled band as in the main FSPO clipping rule, so that both the primary and dual clipping remain consistent with the length-fairness principle.

---

### Official Review · Reviewer_FCnS · 2025-10-31

**Soundness:** 3
**Presentation:** 3
**Contribution:** 2
**Rating:** 6
**Confidence:** 3

**Summary:**

This paper proposes FSPO (Fair Sequence Policy Optimization), a novel sequence-level reinforcement learning method for large language models (LLMs) that addresses length bias in importance sampling (IS) weight clipping. The authors identify that fixed clipping ranges in existing methods like PPO/GRPO lead to systematic reweighting of short versus long responses, distorting optimization. FSPO introduces a √L-scaled clipping band on the sequence log-IS ratio to enforce length fairness, formalized through a Length Reweighting Error (LRE) metric. Theoretically, small LRE ensures a cosine directional guarantee between clipped and true updates. Empirically, FSPO is evaluated on mathematical reasoning tasks using Qwen3 models, showing improved performance over baselines on benchmarks like MATH500 and AIME24/25, with flattened clip rates across length bins and stabilized training.

**Strengths:**

- Novelty: FSPO is the first method to explicitly address length bias in sequence-level IS clipping, filling a critical gap in RLVR literature.
- Theoretical Foundation: The LRE metric and cosine guarantee provide a rigorous basis for length fairness, supported by Markov chain CLT.
- Empirical Validation: Comprehensive experiments on math benchmarks show consistent improvements, with strong ablation studies and diagnostics.
- Practicality: FSPO is easy to implement and compatible with existing RL components, making it accessible for real-world applications.
- Reproducibility: Detailed configurations and open-source references facilitate replication.

**Weaknesses:**

- Limited Scope: Experiments focus solely on mathematical reasoning tasks; generalization to other domains (e.g., code generation or tool-use) is not verified.
- Compute Constraints: Hyperparameter tuning is limited due to resource costs, potentially affecting optimality across diverse settings.
- Assumption Dependency: Theoretical guarantees rely on assumptions like bounded stratification, which may not hold in all scenarios.
- Empirical Drift Simplification: Setting μ≈0 for drift terms is justified empirically but might not generalize to policies with large KL divergence.

**Questions:**

- How does FSPO perform on non-mathematical tasks, such as code generation or dialogue, where length distributions may differ?
- Could the √L scaling be adapted dynamically based on task-specific length variance, rather than using a fixed σ estimate?
- What are the implications of the bounded correlation assumption (Assumption 3.2) in practice? Are there cases where it might fail?
- How sensitive is FSPO to the choice of c (clip scale) across different model architectures or reward functions?
- Have the authors considered combining FSPO with advanced advantage estimators for further gains?

---

> ### Author Response · Authors · 2025-11-21
>
> Thank you for your thoughtful feedback!
>
> **Update of paper:**
>
> We have updated our paper to include an Adaptive FSPO variant (Section 7.5), which dynamically estimates the clipping scale using a running average estimator, eliminating the need for pilot runs for hyperparameter tuning while preserving performance.
>
> **Response to Weakness 1 & Question 1:**
>
> We agree that our experiments are currently limited to RLVR tasks in mathematical reasoning. We chose math because it is currently the most prominent and challenging domain for RLVR, and it intrinsically exhibits diversity in both content and length distributions. Within this domain, FSPO makes a strong contribution not only in terms of performance but also in the effective use and stabilization of response length.
>
> That said, we believe FSPO has promising potential for transferability. The method is conceptually simple and easy to implement, and it provides concrete insight into how to tune clipping when moving to new tasks. One can follow the same recipe as in our main experiments: use a short pilot run to estimate $\hat{\sigma}$ and then set the base scale $c$. Alternatively, as described in Section 7.5 of the updated version, the adaptive variant estimates the clip band online and eliminates the need for pilot run. In both cases, the clip range can adapt to the task-specific distribution of sequence log-IS ratios and lengths. Empirical evaluation of FSPO on other tasks is an important direction, and we leave it as future work.
>
> **Response to Weakness 2 & Questions 2 and 4:**
>
> As mentioned in the above, we use a pilot run to obtain a global estimate of $\hat{\sigma}$ in our main experiment. The same recipe can be applied to other tasks.
> Moreover, we agree with the reviewer that pilot runs are still compute overhead. Motivated by this feedback, the updated version includes an adaptive variant that tracks $\hat{\sigma}$ online (via an EMA of the batch-wise standard deviation of the log-IS ratio) and updates the clip band accordingly throughout training. This directly answers Q2: the $\sqrt{L}$ scaling is made dynamic via $\hat{\sigma}_t$, which automatically adapts to task-specific length variance without requiring estimate from a separate run. Empirically, this adaptive FSPO achieves comparable performance.
>
> **Response to Weakness 3 & Question 3:**
>
> For Assumption 3.1, we view it as an intuitive requirement for any reasonable clipping mechanism: as long as clipping is not too aggressive and only a small fraction of extreme outliers are clipped, the gradient norm **in expectation** after clipping should remain close to that of the unclipped gradient.
> For Assumption 3.2, we agree that there is room to refine this condition. However, we wanted to emphasize that the conclusion of Theorem 3.1 itself does not rely on this assumption being “tight”: when the Length Reweighting Error (LRE) is small, i.e., $|q(L) - \bar{q}|$ is nearly constant across lengths, the correlation between $|q(L) - \bar{q}|$ and $|g_L^\star|$ is naturally small and the factor $\gamma$ is close to 1. When LRE is large, the mildness of this correlation may indeed fail, and the bound in Assumption 3.2 can become looser (with a larger $\gamma$); this simply reflects that, for large LRE, the cosine lower bound in Theorem 3.1 may deteriorate. In that sense, the current form of Assumption 3.2 is consistent with the theorem: it certifies strong alignment only when LRE is small, and allows the bound to be weaker when LRE is large. We may explore refined formulations of this assumption, but the current one already matches the intended interpretation of Theorem 3.1.

---

> ### Author Response · Authors · 2025-11-21
>
> **Response to Weakness 4:**
>
> We appreciate this insightful observation. Indeed, the empirical choice $\hat{\mu} \approx 0$ is specific to our current experimental setting and will not necessarily hold in all scenarios. We emphasize, however, that FSPO does not fundamentally require $\mu = 0$; setting $\hat{\mu} = 0$ is simply a convenient choice in our experiments where the estimated drift is small relative to $\hat{\sigma}$. We expect FSPO to remain applicable and potentially even more beneficial in regimes where $\mu \neq 0$.
> In fact, a very recent work GRPO-guard (https://www.arxiv.org/abs/2510.22319, we have added discussion and reference of it in the Method section in our updated paper) applying GRPO to flow-matching for image generation reports a systematic bias of the sequence-level IS ratio below 1 (which corresponds to $\mu < 0$ for our log-IS). That work observes that this drift induces asymmetric clipping: GRPO’s clipping effectively becomes looser for positive-weight samples, leading to over-optimization, and they propose normalizing the IS weights as a remedy. FSPO’s perspective on the distribution of sequence log-IS ratios is actually well suited for this situation: it explicitly encourages practitioners to observe the empirical distribution of the log-IS ratio and to adjust the clip band dynamically (e.g., via a pilot run or an adaptive running estimate similar to $\hat\sigma$) so that clipping remains symmetric and well-calibrated. This can address the asymmetry issue while preserving the IS semantics (unlike directly normalizing the ratios). Note that symmetric clipping is not the original primary motivation of FSPO: our main goal is to balance clip fractions across lengths, but in the presence of nonzero drift it becomes a useful side effect that fixed clip ranges in baseline methods cannot provide.
>
> **Response to Question 5:**
>
> Yes, we have! One of FSPO’s main advantages is its simplicity and modularity. FSPO only modifies the clipping band for sequence-level IS and does not rely on stacking multiple tricks. As a result, it is orthogonal to many existing techniques in advantage estimation, data filtering, and reward shaping, and can be combined with them in a straightforward way. In our experiments, we deliberately avoid adding additional advanced components so that the improvements can be attributed cleanly to FSPO itself. Nevertheless, incorporating more sophisticated RL components would be natural.

---

### Official Review · Reviewer_UsBx · 2025-10-31

**Soundness:** 3
**Presentation:** 3
**Contribution:** 3
**Rating:** 4
**Confidence:** 3

**Summary:**

This paper proposes FSPO (Fair Sequence Policy Optimization), a sequence-level reinforcement learning method for large language models (LLMs) that addresses the issue of length bias in importance sampling (IS) weight clipping. The authors identify that fixed clipping ranges in existing sequence-level RL methods (e.g., RLOO, GSPO) disproportionately affect sequences of different lengths, leading to unstable training and suboptimal performance. FSPO introduces a length-scaled clipping mechanism. Theoretical analysis formalizes this via Length Reweighting Error (LRE), linking small LRE to directional fidelity in policy updates. Empirical results on mathematical reasoning tasks (MATH500, AIME24/25) demonstrate that FSPO stabilizes training, flattens clip rates across lengths, and outperforms baselines, especially on larger models (e.g., Qwen3-8B).

**Strengths:**

1. The paper clearly articulates a critical underexplored issue—length-dependent bias in sequence-level RL clipping—and formalizes it through LRE, providing theoretical grounding.

2. FSPO is a simple yet effective modification to existing methods, requiring minimal changes (e.g., plug-in log-space clipping) while maintaining compatibility with RL frameworks like GRPO.

3. Experiments are comprehensive, covering multiple model sizes (1.7B/8B), benchmarks, and baselines. Diagnostic plots (e.g., clip fraction vs. length) convincingly validate the method’s fairness claims.

4. FSPO achieves consistent gains, with notable improvements on harder tasks (AIME) and larger models, suggesting scalability and practical utility.

**Weaknesses:**

1. The method relies on a tuned scaling factor, which may require pilot runs for new settings. The paper notes compute constraints limited hyperparameter search, raising questions about robustness.

2. Baseline Comparisons: While FSPO outperforms RLOO/GSPO, ablations show that simply widening the clip range fails, but more analysis on why FSPO's scaling is optimal is needed.

**Questions:**

see weakness

---

> ### Author Response · Authors · 2025-11-21
>
> Thank you for your thoughtful feedback!
>
> **Update of paper:**
>
> We have updated our paper to include an Adaptive FSPO variant (Section 7.5), which dynamically estimates the clipping scale using a running average estimator, eliminating the need for pilot runs for hyperparameter tuning while preserving performance.
>
> **Response to Weakness 1:**
>
> For the first concern, we indeed did not perform extensive hyperparameter tuning due to compute limitations. However, we would like to emphasize that FSPO still offers better guidance for transferability than existing RLVR baselines. Methods such as GRPO and DAPO expose only fixed heuristic clip ranges (e.g., 0.28 / 0.20) and reuse these values across different settings without principled instructions for retuning. In contrast, FSPO sets the base clipping scale (c) from the empirically observed per-length standard deviation $\hat{\sigma}$ of the sequence log-IS ratio. Although estimating $\hat{\sigma}$ requires pilot run, this procedure provides explicit guidance on how to choose (c) when transferring to new domains.
>
> Moreover, to further reduce the cost of such pilot runs, in the updated version we introduce an adaptive variant that estimates $\hat{\sigma}$ using a running average estimator and adjusts the clip range online. This adaptive FSPO is implemented and empirically validated in Section 7.5, and it further reduces the hyperparameter-tuning overhead while preserving performance, showing FSPO's potential for adaptively maintaining clip fairness in different distributions.
>
>
> **Response to Weakness 2:**
>
> We would like to clarify that we do not perform ablations that decompose FSPO into trick components, because FSPO is intentionally minimal: it only modifies the clipping mechanism in the sequence-level RL setting. As described in Section 6.3, we deliberately control the baseline methods and avoid additional heuristics such as prompt filtering, so that RLOO and GSPO already serve as proper ablations for evaluating FSPO.
>
> The ablation in Section 7.4 instead addresses a natural question: since FSPO’s clip band scales as $b_L = c \sqrt{L}$, could the improvement simply come from being more permissive to long sequences?
> To isolate this factor, we implement a baseline that enlarges a fixed clip band (not scaled by $\sqrt{L}$), making longer sequences more likely to be accepted but without length balancing. This variant performs worse than even the original baselines, showing that FSPO’s improvement is not simply due to being more permissive on long sequences. Instead, the improvement arises from balancing the clip fraction across lengths, ensuring a correct update direction.

---

### Meta-Review · Area_Chair_gmZU · 2025-12-12

**Summary:**

This work proposes a new method to clip the ratio in PPO based on the length of LLM responses. Theoretical analysis is provided to justify the proposed approach. Experiments are conducted on math domain and some improvements are observed. My major concern is the empirical improvement. I understand LLM experiments can take long time to run. But it seems all the reported results in the submission is based on a single run with a single seed. And the reported improvements are very very marginal. I am not sure if we can draw any statistically significant conclusion from the reported numbers. This concern is further amplified by the fact that the paper studies only math domain. After all, the experiments are done with 1.7B and 8B models. It is not impossible to provide more runs to gather statistically significant results. I, therefore, recommend reject.

Furthermore, my suggestion is that if the authors want to study the clipping problem from a scientific perspective (instead of just getting a higher number on math tasks), they can consider adding some small scale non-LLM experiments. If the clipping ratio is a problem, it should be a universe problem not just for LLM. Adding some small scale experiments (e.g., some carefully designed grid world) can isolate the problem better and provide finer diagnosis.

**Reviewer Concerns:**

Many reviewers raise concern about the limited evaluation domain. I do not think this is addressed. Many also raised concern about hyper parameter tuning. This may be addressed to some extent by the rebuttals.

**Reviewer Scores:**

UsBx may remain 4 and the rest 3 reviewers may remain 6.

---

### Decision · Program_Chairs · 2026-01-26

Reject